# Feasibility of Electrodermal Activity and Photoplethysmography Data Acquisition at the Foot Using a Sock Form Factor

**DOI:** 10.3390/s23020620

**Published:** 2023-01-05

**Authors:** Afonso Fortes Ferreira, Hugo Plácido da Silva, Helena Alves, Nuno Marques, Ana Fred

**Affiliations:** 1Instituto Superior Técnico (IST), Av. Rovisco Pais n. 1, 1049-001 Lisboa, Portugal; 2Instituto de Telecomunicações (IT), Av. Rovisco Pais n. 1, Torre Norte—Piso 10, 1049-001 Lisboa, Portugal; 3Instituto de Engenharia de Sistemas e Computadores-Microsistemas e Nanotecnologias (INESC-MN), Rua Alves Redol 9, 1000-019 Lisboa, Portugal; 4Meia Mania Lda, Zona Industrial dos Matinhos Pav. 4/5, 3200-100 Lousã, Portugal

**Keywords:** electrodermal activity, galvanic skin response, photoplethysmography, conductive textiles

## Abstract

Wearable devices have been shown to play an important role in disease prevention and health management, through the multimodal acquisition of peripheral biosignals. However, many of these wearables are exposed, limiting their long-term acceptability by some user groups. To overcome this, a wearable smart sock integrating a PPG sensor and an EDA sensor with textile electrodes was developed. Using the smart sock, EDA and PPG measurements at the foot/ankle were performed in test populations of 19 and 15 subjects, respectively. Both measurements were validated by simultaneously recording the same signals with a standard device at the hand. For the EDA measurements, Pearson correlations of up to 0.95 were obtained for the SCL component, and a mean consensus of 69% for peaks detected in the two locations was obtained. As for the PPG measurements, after fine-tuning the automatic detection of systolic peaks, the index finger and ankle, accuracies of 99.46% and 87.85% were obtained, respectively. Moreover, an HR estimation error of 17.40±14.80 Beats-Per-Minute (BPM) was obtained. Overall, the results support the feasibility of
this wearable form factor for unobtrusive EDA and PPG monitoring.

## 1. Introduction

Psychophysiological data acquisition with wearables has been shown to play an important role in the ongoing medical paradigm shift, i.e., moving from disease treatment to prevention and health management [1]. Using neuropsychiatry as an example, early research has demonstrated that wearables have the potential to accurately monitor the health condition of patients with major depression [2], bipolar disorder [3], and epilepsy [4]. Often, health monitoring through wearables can even represent an adjunct therapeutic plan that extends established practices.

The impact of wearables is even more significant when considering the prevalence of some of these diseases. Neuropsychiatric disorders are considered to be the leading cause of disability and the second leading cause of death worldwide [5]. According to the Institute for Health Metrics and Evaluation (IHME), it is estimated that around 2.6 billion people (35.7%) and 970 million (13%) worldwide suffer from neurological and mental disorders, respectively [6].

Regardless of the clinical application, and despite the foreseen potential of wearables for health management, there are evident barriers limiting their acceptability by patients, chief among which are user privacy, discretion, and practicality. Patients have reported that bulky and exposed devices would increase their condition awareness and feelings of anxiety, fearing social stigma and discrimination as a result [7]. Consequently, there is a strong preference for aesthetic, unobtrusive, and ergonomic form factors of wearable devices [8].

While hand- and wrist-worn sensors are acceptable form factors for fitness and general wellbeing applications, for patients, more discreet options are needed. As such, in this paper, the state-of-the-art is extended by exploring the feasibility of multimodal physiological data acquisition at the foot/ankle region, by using electronic textiles (e-textiless) as the skin-sensor interface at the foot, and by integrating all the components in a sock form factor that can be deployed at scale. With this approach, the ongoing research towards long-term, unobtrusive, and comfortable physiological monitoring is further explored.

The set of sensors most commonly found in wearables consist of an accelerometer, a Photoplethysmography (PPG) sensor, and an Electrodermal Activity (EDA) sensor [3,4], which are used to track body movement, Heart-rate (HR) and autonomic arousal, respectively. Multimodal data acquisition has been shown to outperform unimodal approaches and to provide promising results in multiple clinical applications [2,3,4,9]. In fact, there are even FDA-approved commercially available wearable devices that use the aforementioned combination of sensors (e.g., Empatica Embrace for epileptic seizure monitoring).

Most of the work found to date concerning the use of PPG and EDA sensors have focused on the upper limbs, mainly the hands and wrists. However, it is suggested that more remote body locations, such as the feet, could also be used to record PPG [10] and EDA [11,12,13,14] signals with comparable quality. Limited research has been conducted on the latter, the reason for which, in this work, the feasibility of Acceleration (ACC), PPG, and EDA data acquisition in a sock form factor are studied, along with the possibility of using e-textiless as the skin-sensor interface. Due to the foreseen importance of PPG and EDA signals in discriminating health states in patients’ neuropsychiatric disorders, throughout this work, special focus is given to these modalities.

## 2. Background

### 2.1. Electrodermal Activity

Sweating is a physiological phenomenon mediated by the sympathetic branch of the Autonomic Nervous System (ANS), and occurs at the eccrine glands, which are scattered through most of the body, although most numerously on the soles of the feet and palms (∼600 to 700 glands/cm${}^{2}$) [15]. EDA is the biosignal that reflects changes in the amount of sweat that is excreted at the skin surface, usually expressed as conductance; due to its typical amplitude range, the unit for EDA is often expressed in microsiemens [μS] [16].

Apart from thermoregulatory changes in sudomotor activity, it was experimentally confirmed that changes in EDA also occur as a result of both immediate and long-term emotional arousal (also referred to as “emotional sweating”) [17]. Additionally, differences in EDA signals have been observed in patients with multiple clinical conditions, when compared to healthy individuals [18,19,20]. Therefore, EDA also reflects specific pathological states mediated by the ANS [21,22].

EDA signals have two main components: the Skin Conductance Level (SCL) and the Skin Conductance Response (SCR). The SCL, consisting of low-frequency oscillations, is often described as the background or tonic component. The SCL is prone to inter-subject variability, but may also vary significantly for the same subject when under different psychological states [23]; typical amplitudes range between [2–20] μS [23]. The SCR, consists of high-frequency oscillations with low amplitude usually occurring shortly after a stimulus (e.g., a deep breath or a bodily movement). When a stimulus is wittingly presented, the SCR occurring shortly afterward is referred to as specific SCR. Otherwise, it is considered non-specific Skin Conductance Response (NS-SCR) [23].

The amplitude threshold for identifying specific and non-specific SCR events is usually between 0.1 and 0.05
μS [24]. To distinguish SCR events, it is common to define a window in time such that if a SCR occurs after a stimulus is presented within this window, it is considered a specific SCR. Usually, a window between 1–3 or 1–4 s is considered [23]. When studying and comparing EDA signals, SCR features are often considered, which include its amplitude, latency (i.e., the time between the stimulus and the SCR onset), and the recovery time, as shown in Figure 1.

The gold standard for EDA acquisition in laboratory research consists of applying two electrodes on one hand, preferably the non-dominant one, to avoid motion artifacts. Typically pre-gelled electrodes made of Silver/Silver Chloride (Ag/AgCl) are placed at the volar surfaces of the index and middle fingers. However, the electrodes can also be placed at the thenar and hypothenar eminences of the palms [25].

### 2.2. Photoplethysmography

Photoplethysmography (PPG) is an optical method to measure variations of blood perfusion in the superficial vascular tissue bed [26]. It modulates the cardiovascular cycles and is mostly used to extract HR-related features. Experimental research using the PPG has demonstrated that this method can be used to estimate HR with comparable performance to that of Electrocardiography (ECG), in both healthy subjects and patients at risk of developing cardiovascular diseases [27,28], with the advantage of requiring the application of the sensor in a single anatomical location.

The simplest PPG sensors have one Light Emitter (LE); usually, a LED, emitting within the wavelength range of 600–900 nm (i.e., from green up to near-infrared) and a Photodetector (PD), such as a photodiode. The locations of the LE and PD can either be on the same side of a given body part (i.e., the fingertip) or on opposite sides, for which the PPG is considered to be on reflective-mode or absorbance-mode, respectively. In this work, only the first type is considered because the emitted light is not expected to traverse large sections of the body as those found in the lower limbs.

When a reflective-mode PPG sensor is placed in close contact with the skin, the incident light reaches the blood vessels, and part of it is backscattered and reaches the PD. Upon increases in the local blood volume, the incident light is absorbed by blood constituents (mainly water); therefore, the amount of light reaching the PD decreases.

The typical PPG waveform contains two wave components. The DC (quasi-static) component varies slowly with time and depends on the average blood volume and respiration motion, whereas the AC (pulsatile) component is given by oscillations that are superimposed on the Direct Current (DC) component and have a frequency that matches the HR. After proper signal amplification and filtering, a PPG signal exhibits a typical morphology containing the systolic peak, the dicrotic notch, and the diastolic peak, as shown in Figure 2.

## 3. Related Work

Previous research has already demonstrated a high correlation between simultaneously acquired EDA signals at the hand and foot in healthy individuals [11,12,13,14]. However, few experimental research actually focused on a sock form factor and/or using e-textiless. One research project called MONARCA aimed at investigating the feasibility of long-term EDA monitoring in patients with bipolar disorder, and proposed to integrate an acquisition system in a wearable sock for this purpose, although, so far, the authors have not confirmed the use of textile EDA electrodes in this project.

In one of their papers, EDA signal acquisitions were performed simultaneously at the hand and foot in 8 healthy subjects [11]. The subjects were asked to take 3-s deep breaths to compare induced ANS-mediated SCRs at the hand and feet. During the experiment, the participants also mimicked light limb movements to address motion artifacts. The 3-s deep breath stimuli elicited SCRs 82.4% of the times at the hand and 79.9% at the foot. Moreover, the authors computed the consensus of peaks between the two locations. For this, they first defined a time window of 4.0 s such that, for every peak in the hand (reference), if one peak at the foot occurred within that window (i.e., −2.0 to 2.0 s with regard to the hand peak), it would be counted as a co-occurring peak. If the windows of two consecutive reference peaks overlapped and one event at the foot were within the intersection, then the windows would be shortened until the foot peak was only counted in one of the windows to avoid double-counting. Finally, if two foot-peaks were within the same window, this would not be counted as co-occurring. The consensus value would then be given by the ratio between the number of co-occurring peaks and the total number of peaks detected at the hand. For all subjects, the authors obtained a consensus of 88% for specific SCRs and 50% for NS-SCRs.

As for the effect of body movement, the authors concluded slight limb movement did not have a significant effect on the distribution of both evoked SCRs and NS-SCRs, likely because the pressure contact between the electrodes and the skin remained constant. The authors concluded their prototype is capable of measuring relevant EDA features at the feet of healthy individuals.

In another paper related to the same project, Gravenhorst et al. [29] performed EDA acquisitions at the medial arch of the foot in 11 patients diagnosed with bipolar disorder. In this experiment, researchers detail an improved acquisition setup with a conductance measurement range of 0.2–100 μS and a sampling frequency of 51.2 Hz. The authors used “sticky electrodes” which were hidden under a sock. Each subject would attend a session of conversation with the doctor for 30 min (talking condition), followed by a session in which they listened to the oddball paradigm stimulus (oddball condition). Researchers used a triaxial accelerometer to check and discard segments of recordings corrupted with motion artifacts. Then, various statistical features were extracted from the EDA signals to assess if the two conditions could be discriminated. The authors concluded that 12 features proved to be statistically significant for the two considered conditions, with the number of maxima in the 2nd derivate of the EDA signal being the most significant one. The authors highlighted the technical feasibility of their EDA acquisition interface for the feet.

Healey [30] developed an EDA-monitoring sock with textile electrodes using an acquisition device with 12-bit resolution. First, simultaneous acquisitions were performed with the sock together with standard electrodes on the same foot. As for the textile electrodes, these were placed at the ball and heel of the foot. Several hours were recorded under very different conditions, such as walking, standing, and sitting at a desk, but the number of tested subjects was not disclosed. The collected data has shown a high correlation (p<0.001) for the sitting condition only. The authors also performed simultaneous acquisitions using the sock compared with standard pre-gelled electrodes placed at the hand, and, although no statistical analysis was made, the collected signal exhibited a similar morphology. The authors concluded their EDA-monitoring sock matches the palm-based system and could hence be used in the future to study the detection of arousal states.

Although PPG is typically acquired at the upper limbs (e.g., finger, wrist, and arm), chest, and head [31,32], some research has provided validation of foot-based PPG acquisition.

Hong and Park [33] have identified regions of the sole suitable for PPG acquisition using an optimized array of sensors (i.e., multiple pairs of LED and PD) matching the near-infrared region of the light spectrum at 890 nm. In this work, researchers tested various foot locations in subjects in standing and steady positions. For this purpose, simultaneous PPG acquisitions were performed at the finger (i.e., reference) and multiple testing locations within the foot, using a sampling frequency of 500 Hz. Then, well-known signal quality indices [34] were used to assess the resulting acquisition performances. It was observed that, within an experiment involving 53 subjects, the highest signal quality was obtained from PPG sensors located at foot regions near the lateral plantar artery, and plantar arch between the anterior longitudinal sulcus and the heel. In fact, a total mean HR error of 0.638 BPM was obtained from 12 sensors located within the aforementioned regions of the sole, for all 53 subjects. The authors suggest regions having large blood vessels accounted for higher signal-to-noise ratios and, thus, increased similarities with the reference finger PPG. Finally, the authors concluded that no significant differences were observed for the HR error or any of the signal quality indices when the subject recordings were split by age or gender.

In another paper, Jarchi and Casson [10] have tried to validate PPG recorded at the ankle during cycling exercises in 8 subjects. PPG and simultaneous triaxial accelerometer signals were recorded at the wrist and the ankle using a sampling frequency of 256 Hz, along with ECG recorded at the chest (i.e., serving as a ground-truth signal). The authors showed that for resting conditions, it was possible to obtain clear signals without applying sophisticated signal processing. As for fast cycling exercises, the authors obtained a root mean squared error of the HR of 9.1±3.1 BPM (mean and standard deviation of the metric, comparatively with the ECG recordings), after applying their own signal processing algorithm. The authors conclude that the acquisition performance was satisfactory considering the high susceptibility of the ankle-PPG to motion artifacts.

Overall, previous work has already demonstrated the feasibility of using standard EDA electrodes at the feet with comparable results as using the same electrodes at the hands, considering both healthy individuals and patients with one neuropsychiatric disease. However, to the best of our knowledge, in only one paper, researchers designed and tested a prototype sock that actually integrates textile EDA electrodes [30]. Nonetheless, in this paper, no information was disclosed regarding the number of tested subjects, and the analysis of the acquired signals was rather simplistic. Moreover, there is growing evidence that HR features could be successfully extracted from PPG recorded at the lower limbs, although this location is prone to motion artifacts, especially during intense physical exercise.

To the best of our knowledge, our work provides the first in-depth comparative analysis of EDA signals acquired using textile electrodes at the foot and standard pre-gelled electrodes at the hand in a well-defined population of healthy subjects. Our work also contributes to ongoing research in the acquisition of peripheral biosignals, by testing the performance of HR estimation from PPG signals measured at the ankle. Moreover, in our approach, a wearable smart sock for the multimodal signal acquisition was developed, providing a description of a simplified prototyping method for the integration of textile EDA electrodes in garments. Our approach confirms the feasibility of using conductive textiles as EDA electrodes and provides future directions on the development of customized EDA and PPG sensors for biosignal acquisition at the ankle/foot.

## 4. Methodology

### 4.1. System Overview

The system consists of a wearable sock with integrated EDA electrodes to which a data acquisition device (herein referred to as the main unit) connects at the ankle region. As shown in Figure 3, the main unit contains the ScientISST CORE, which is the acquisition device, the PPG sensor and the accelerometer, which are internally connected, and a small battery for power supply. Moreover, the main unit has a hole especially designed to fit the PPG sensor, and the EDA and temperature sensors are externally connected, allowing to adapt the EDA sensor more easily without having to open the main unit.

The ScientISST CORE [35] is a novel signal acquisition board especially developed for biomedical applications, with an integrated Bluetooth module for wireless transmission of the recorded signals. It has a controllable sampling rate, $f_{s}$, although 1 kHz was used in the experiments in this work, and has an Analog-to-Digital Converter (ADC) resolution of 12-bit [35].

### 4.2. Biosignal Sensors

As an add-on to our acquisition setup, the BITalino triaxial accelerometer (PLUX, S.A., Lisbon, Portugal) was included in the main unit and placed at the ankle, enabling the detection of possible motion artifacts during our acquisitions. Its transfer function is given by Equation (Equation 1), in which ADC corresponds to the raw data being measured. Moreover, $ADC_{max}$ and $ADC_{min}$ represent the maximum and minimum values derived from the raw data, respectively, registered during a full revolution of the sensor in each axis. Experimentally, $ADC_{min}=541$ and $ADC_{max}=965$ were obtained.
(1)$$ACC=2\times \frac{ADC-ADC_{min}}{ADC_{max}-ADC_{min}}-1\;\;\left[g\right]$$

Multiple research has already demonstrated evidence of the influence of body temperature on the morphology of the EDA signals recorded from the hands [36]. Thus, a temperature sensor (TMP36 temperature sensor from Analog Devices, Boston, MA, USA) was used to assess the potential effects of the foot temperature on the acquired signals. According to the datasheet, the sensor has an output voltage scaling factor of 100 mV/${}^{\circ }$C and an offset voltage of 0.5 V; thus, its transfer function is given by Equation (Equation 2), where *n* represents the resolution of the acquisition device (in bits).
(2)$$T=\left(\frac{ADC\times V_{cc}}{2^{n}}-0.5\right)\times 100\;{[}^{\circ }C]$$

For the two main biosignals of interest, PPG and EDA sensors were used. An off-the-shelf reflective-mode sensor was used for PPG acquisition (Pulse Sensor from World Famous Electronics LLC, New York, NY, USA). This sensor is placed at the ankle, as this location allows a suitable integration of the sensor in the sock and has been reported as a suitable anatomical location for PPG acquisitions [10]. For the EDA acquisition, the BITalino EDA 081217 sensor (from PLUX, S.A., Lisbon, Portugal) was considered; this sensor works under the DC exosomatic measurement method, has been used in multiple research projects [37,38,39] and has a conductance measurement range of [0,25]
μS.

### 4.3. Design and Development

To use conductive textiles as EDA electrodes, it is necessary to first define the anatomical location within the foot for EDA acquisition. As seen in Section 3, the medial arch is a well-validated foot location for this purpose. Additionally, although toes I and II replicate more closely the standard electrode application at the hand, these locations have a higher degree of freedom when walking compared to the medial arch, potentially being more prone to motion artifacts. For example, the toes have the ability to undergo dorsiflexion, which is a motion relative to the rest of the foot [40]. As such, the medial arch was chosen in the scope of this work; in addition to being a more stable anatomical location, it represents a larger area of the foot, and conventional socks already provide contact between the fabric and the skin within this location.

Figure 4 shows some of the main steps for the integration of the EDA electrodes in the sock. First, the region corresponding to the medial arch was drawn, as shown in Figure 4a. The socks were cut along the sagittal plane of the foot, and then the lycra was assembled into the half containing the medial arch region. Two patches of lycra served both for signal conduction and as electrodes (as shown in Figure 4b), and were cut and fixed to the fabric using iron-on-adhesive, which is shown in Figure 4c. Then, the electrodes were passed to the inner side of the sock and fixed, as shown in Figure 4d. To further secure the conductive segments, these were sewn using a sewing machine. The two ends of the patches near the ankle were crimped to a connector so that the EDA sensor could be easily connected to the resulting sock.

After assembling the EDA electrodes, a flexible connector was 3D printed using a flexible polymer and sewn to the sock at the ankle region, where the main unit can be attached. This connector has a small opening to fit the PPG sensor of the main unit, giving direct access to the skin surface of the ankle. The prototyped smart sock is shown in Figure 5.

The electrical resistances across the conductive segments were measured. Since the resistance of the conductive materials was expected to be considerably higher than the cables used for resistance measurements, the two-wire method was performed. For this, a digital multimeter Agilent 3458A (from Agilent Technologies, Santa Clara, CA, USA), was used. For each segment, one cable was connected to the EDA electrode, and the other was connected to the wire of the EDA sensor, to take into account possible additional resistance contributions, namely, contact resistance between the conductive lycra and the sock’s connector. Figure 6a shows the setup used to measure the electrical resistance. Two measurements were performed, one for each conductive segment, since slightly different conductive paths (i.e., the path length and the shape of the conductive material) are likely to have different resistances. Moreover, a total of 6 measurements were performed for each conductive segment.

As shown in Figure 6b, the EDA electrodes consist of two rectangles spaced 5 ± 0.5 mm apart with dimensions (15×10) ± 0.5 mm (where the last value represents the measurement error of a standard ruler). It is worth mentioning that preliminary research conducted in this work has revealed that the aforementioned electrode shape and dimensions, as well as inter-electrode distance, allowed for successfully recording EDA signals at the foot using the same type of conductive lycra.

### 4.4. Protocol

In the experimental procedure for signal acquisition, EDA and PPG signals were simultaneously recorded using the sock and a reference device. As a reference device, the BITalino (r)evolution Plugged (PLUX, S.A., Lisbon, Portugal) was used, which was connected to the same type of sensors. It is worth mentioning that the BITalino has been widely used within the research community for EDA acquisitions using the BITalino EDA sensor (as used in this experiment) [37,38,39,41]. Furthermore, this device has also been used for PPG acquisitions in laboratory research contexts [42,43]. For these reasons, the BITalino was selected as the gold standard device for reference data acquisitions in this study.

To accurately synchronize the two acquisition devices (i.e., the smart sock and the BITalino), optical synchronization was used, thus, ensuring electrical decoupling between both devices. For this, the main unit of the smart sock was connected to a light sensor for the visible spectrum and the BITalino to a LED, also operating in the visible range. Respectively, these acquisition devices record the measured light of the light sensor and the digital output of the LED, so that the signals can be synchronized in post-processing.

To minimize the time overhead for the participants and perform a more accurate evaluation of each sensor, either EDA or PPG data were acquired in each session. For subjects in which the EDA modality was considered, simultaneous EDA signals were acquired at the hand (reference location) and foot (testing location), in addition to foot acceleration and temperature signals. For these acquisitions, the subjects were instructed to fill out a simple questionnaire to assess their anthropometric data, their tendency to sweat from the hands and feet, and sensorial perception of their body at the time of the experiment, as shown in Figure 7. The experimental setup for this experiment is shown in Figure 8 with the main labeled parts. Each subject first wore the sensorized sock (1) on the right foot with the main unit (2), after which the EDA sensor of the device was connected to the sock. The temperature sensor (3) was then connected. This sensor was placed under the sock’s fabric and superiorly to the metatarsus. Then, the BITalino device (4) was placed on a non-conductive surface. The light sensor (5) was coupled with the LED (6). Finally, for the reference EDA acquisition, pre-gelled electrodes made of Ag/AgCl embedded in a conductive hydrogel (7) were placed at fingers I and II of the right hand. These electrodes are the Kendall H124SG ECG electrodes (from CardinalHealth, Dublin, OH, USA).

For the PPG acquisition, the setup is similar, albeit more simple; instead of the EDA electrodes, the subject wore the pulse sensor at the index finger (i.e., at the distal phalange of this finger, on the right hand) which is connected to the BITalino device. In both the experimental sessions for EDA and PPG acquisitions, data were recorded for 300 s, and subjects were instructed to remain as still as possible during the experiment, especially at the hands and feet, to avoid motion artifacts.

### 4.5. Signal Processing

After synchronizing the hand and foot recordings, EDA signals were filtered using a fourth-order Butterworth low-pass filter with a cut-off frequency, $f_{c}=5$ Hz, and then smoothed using a moving average filter with a window size of 750 ms, obtaining $EDA_{filt.}$. This filtering step was performed using the BioSPPy toolbox [44]. Furthermore, from $EDA_{filt.}$ signals, the SCRs component was extracted by applying the cvxEDA algorithm [45], obtaining $SCR_{cvxEDA}$. The temperature signal was filtered using a moving average filter with a window of 2 s. As for the ACC data, a moving average filter with a window of 500 ms was applied to the signals.

As for the PPG recordings, after synchronization, these signals were processed in Python using BioSPPY [44]. The PPG signals were initially filtered with a fourth-order Butterworth band-pass filter with lower ($f_{c_{L}}$) and higher ($f_{c_{H}}$) cutoff frequencies of 1 and 8 Hz, respectively. A robust algorithm [46] was then used to automatically detect systolic peaks from the filtered signals and estimate the instantaneous HR.

## 5. Experimental Results

### 5.1. Validation of EDA Electrodes

Table 1 shows the resulting measured resistances. These values were 32.65±0.82 and 32.06±0.18
Ω for the posterior and anterior electrodes, respectively. Moreover, from all 6 measurements on each segment, the maximum difference in resistance observed was 1.67 Ω, which represents approximately 5.16% of the average resistance of both the electrodes, 32.36 Ω. This average resistance is approximately 30.90 mS, which is far above the [0–20] μS measurement range of the EDA sensor. Since the resistance of the skin accounts for a higher impedance compared to the resistances of the electrodes, apart from contact resistance (i.e., between the electrodes and the skin), it is expected that the measured signals are going to fall within the measurable conductance range of the sensor. Moreover, the difference between the resistance of the two electrodes is minimal.

### 5.2. Population

A total of 34 subjects, 13 males and 17 females with ages above 18 years old, were enrolled in the experiments. Moreover, among the subjects, 19 were assigned to the EDA acquisition and 15 to the PPG acquisition. The anthropometric data are summarized in Table 2 and Table 3 for the EDA and PPG acquisitions, respectively. For subjects in the EDA and PPG groups, the anthropometric parameters height and foot size had significant differences between the male and female subjects; additionally, for subjects within the PPG group, the anthropometric parameter weight also had significant differences between the two genders.

Table 4 summarizes the answers to the questions for the assessment of the tendency to sweat and the sensorial perception of the subjects before the experiment.

### 5.3. EDA Data Analysis

EDA signals were successfully recorded at the hand in all of the 19 subjects; however, in 8 of these subjects, no EDA signal could be measured at the foot (i.e., constant null signal for the full duration), resulting in 11 subjects with measurable foot-EDA signals. Furthermore, a full non-zero EDA signal was only observed in 3 out of these 11 subjects, whereas in the remaining 8, the EDA signals were only non-zero during part of the acquisition.

To understand if there were anthropometric or physiological factors associated with cases of non-measurable EDA, the following variables were first considered: continuous numerical candidate (independent) variables and the ratio of measurable EDA, $R_{EDA}$, given by Equation (Equation 3) (where *S* is the signal). The candidate variables consisted of height, weight, and foot size (from the questionnaire in Figure 7a), and statistical features extracted from ACC and temperature signals (for temperature features, we considered only 15 subjects). The statistical features consisted of the mean, median, variance, standard deviation, skewness, entropy, min, max, range (i.e., max–min), interquartile range, and mean absolute deviation. We computed the Pearson correlation between these candidate variables and $R_{EDA}$. None resulted in a statistically significant correlation (i.e., *p*-value < 0.05).

To test the correlation between age range and $R_{EDA}$, we used the Spearman correlation, which also did not result in a statistically significant correlation. In addition, we tested whether $R_{EDA}$ differed between gender groups or groups answering differently (binary answers) in each of the 5 questions regarding sweating tendency (from the questionnaire in Figure 7b). For this, we used ANOVA and Mann–Whitney U tests when the $R_{EDA}$ was normally and non-normally distributed in the groups, respectively. Again, for these variables, no significant differences were observed.
(3)$$R_{EDA}=\frac{\mathrm{Nr}.\;\mathrm{samples}\;\mathrm{where}\;S>0}{length(S)}$$

From the extracted full hand and foot EDA signals, we computed the Pearson correlations between the two body locations. Figure 9 shows the correlation values for each subject and type of extracted signals ($EDA_{filt.}$ and $SCR_{cvxEDA}$). Considering all subjects, the Pearson correlation for $EDA_{filt.}$ was 0.32±0.58 (mean and standard deviation) within a range of [−0.62;0.95]. In the same order, for $SCR_{cvxEDA}$, we obtained 0.39±0.35 within [−0.01;0.93]. The obtained correlation values are dispersed as a result of a high inter-subject variability regarding the similarity of the signals from the two body locations. For example, for $EDA_{filt.}$, a strong correlation was observed (i.e., >0.75) in 3 of the subjects, whereas in 3 other subjects, a negative value was registered. Moreover, if only the SCR component is considered ($SCR_{cvxEDA}$), generally higher correlations are obtained.

Figure 10 shows the $R_{EDA}$ values for the signals obtained from each subject, along with the corresponding durations of the signals in which they were not saturated. For all subjects, the measurable portions of EDA signals had approximately at least 150 s in total. Considering both Figure 9 and Figure 10, some signals yielding the highest Pearson correlation values preserved approximately the full acquisition time. For instance, for subjects with IDs 2, 20, and 21, in which strong Pearson correlation values were observed for $EDA_{filt.}$, their $R_{EDA}$ was >0.75 (i.e., summed measurable portions had a duration >225 s), thus, confirming the significance of such cases.

It is known that not every SCR event is expressed in all body locations. For instance, one SCR peak arising from a given stimulus can be expressed at the hand and be absent at the foot or vice-versa. Moreover, events elicited by the same stimulus can be expressed with latency in different body locations. For this reason, we studied the similarity of the EDA signals in regions with co-occurring SCR events and minimal lag. For this, we first extracted SCR onsets and peaks from the hand and foot $EDA_{filt.}$ signals, using the algorithm described in [47]. On average, considering all 11 recordings, the foot EDA resulted in 3.63 extra onsets and peaks that were undetected in the hand signals.

Then, similarly as in [11], we defined a time window for co-occurring peaks of 4.0 s such that, for each peak expressed in the hand signal (i.e., reference), the co-occurring peak in the foot signal would be assigned as the closest one within −2.0 to 2.0 s with respect to the reference peak. Thus, if no foot peak was within the defined window, the respective hand peak would be neglected. It is worth mentioning that we considered the hand as the reference body location (i.e., by searching nearby foot peaks) because, on average, there were fewer peaks detected at the hand.

Once the co-occurring peaks were found, segments of hand and foot $SCR_{cvxEDA}$ were extracted from a larger time window around each reference peak. We considered a window of [−2,2.2] s, which, based on testing and visualization of different windows, seemed to include the full waveform of the peaks. Figure 11 shows two of the windows used in the correlation analysis.

Figure 12a shows the statistical distributions of the Pearson correlation values for the obtained $SCR_{cvxEDA}$ segments extracted from the peak windows. The obtained correlation was 0.39±0.57 (mean and standard deviation) within a range of [−0.98;1.0], which is unexpectedly more dispersed compared to the full $SCR_{cvxEDA}$ signals.

Using the same method in [11] (and the same window of 4 s), we computed the consensus of peaks, considering the peaks from the hand signals as the reference events. For all subjects, the consensus of peaks was 0.69±0.22 (mean and standard deviation) within a range of [0.23,1.0].

We also analyzed the lag between events, by computing the time difference between each peak at the hand and the closest peak at the foot, within a maximum window of 4 s (i.e., [−2,2] s). Figure 12b shows the distributions of these time lags. Considering all subjects, the time lags were 0.41±0.70 s. The mean value is positive, meaning the peaks measured at the hand occurred, on average, before the closest peak at the foot, which is in accordance with the expected slower response at the foot.

### 5.4. PPG Data Analysis

Initially, using the filtering step described in Section 4.5 resulted in a poor peak detection performance for the ankle PPG (i.e., a considerable number of false positives), which motivated us to fine-tune this pre-processing step. Thus, we adjusted the frequency band of the filtering step. This required the systolic peaks to be manually labeled for finger and ankle PPG. Afterward, two grid searches were performed to obtain the highest peak detection accuracy for the ankle PPG, considering the $f_{c_{L}}$ and $f_{c_{H}}$ cut-off frequencies of the filtering step as hyper-parameters.

In the first grid search, the following linearly spaced values were considered: 5 values in [0.5,1.5] Hz for $f_{c_{L}}$ and 20 values in [8,20] Hz for $f_{c_{H}}$ (i.e., a total of 100 combinations). To increase granularity, the second grid search was performed in sub-intervals, which resulted in higher accuracy than the first one. The second one was performed as follows: 10 values in [0.5,1.2] Hz for $f_{c_{L}}$ and 30 values in [8.9,25] Hz for $f_{c_{H}}$.

A band-pass frequency range of [0.97,11.12] Hz was empirically found to yield the highest accuracy for ankle PPG, without compromising the performance for the finger PPG. In fact, this same frequency range also resulted in the maximum average accuracies for both body locations. The results are shown in Table 5.

Moreover, for each segment, the HR was estimated based on the predicted PPG peaks. Considering all segments, for the index finger, the difference between the true HR (i.e., computed from manually labeled peaks) and the predicted one was −0.72±1.54 BPM (mean ± standard deviation). For the ankle, this difference was −10.13±13.72 BPM, meaning that the predicted peaks resulted in a significant overestimation of the HR. Additionally, we computed the root mean squared error of the HR in terms of the true and predicted HR estimations at the foot. For this purpose, we employed linear interpolation on the HR values because the algorithm we used computed HR values only at the instances of the PPG peaks. The obtained mean squared error of the HR among all segments was 17.40±14.80 BPM.

To study the association between anthropometric data and the performance of the peak detection for ankle PPG signals, we considered candidate (independent) and target (possibly dependent) variables. The candidate variables consisted of age, gender, height, weight, and foot size, obtained from the questionnaire in Figure 7a. For the target variables, we considered the accuracy, sensitivity, and positive predictive value of the peak detection from the ankle signals, as well as the mean difference between the true HR and the predicted one also for the ankle. For age, Spearman correlations were computed, and for gender, the Mann–Whitney U test was performed (none of the target variables was normally distributed). Moreover, for the remaining continuous candidate variables, Pearson correlations were computed. When testing the null hypotheses, candidate variables age, gender, weight, and BMI resulted in *p*-values <0.05 against sensitivity. These results are shown in Table 6, suggesting the peak detection and HR performance tends to increase with age, weight, and BMI and is higher in the male group.

## 6. Discussion

In this study, the similarity of EDA and PPG signals was simultaneously collected at the medial arch of the foot through a wearable sock using conductive lycra and at the hand through standard electrodes. The electrode materials used (i.e., conductive lycra and pre-gelled electrodes) and the anatomical sites for EDA measurement were varied. While performing additional reference acquisitions on the same foot as the sock’s foot could, in principle, have eliminated the effect of body location, doing so would have resulted in mutual interference due to the application of voltage from multiple sources on the medial arch (considering that the EDA sensor works by the DC exosomatic method). Additionally, it is unclear whether using the foot on the opposite side of the sock’s foot (i.e., mirrored by the sagittal plane) would completely eliminate anatomical variation, as the two feet, like the hand and foot on the same side of the body, are associated with different dermatomes and may, therefore, exhibit different skin responses [48].

In the experiments, an EDA sensor optimized for acquisitions in palmar regions was used, which corresponds to the gold standard location for EDA measurements. While such choice eliminated the sensor design as a confounding factor, one limitation of this approach was the saturation in the lower end of the measurement range when recording EDA signals at the medial arch in some subjects. This is likely related to the resistance of the foot being higher than the resistance of the hand. In fact, in one paper [49], it was found that the medial arch had lower SCL levels comparatively to the fingers, in both resting (baseline) conditions, and during a stress-inducing task. The SCL values observed at the medial arch were 5.56±3.01
μS (mean and standard deviation) at rest and 6.82±4.95
μS during the stressful task.

The resistance of the segments made of conductive lycra (i.e., from the connecting region at the ankle down to the medial arch where it establishes contact with the skin) is negligible compared to the expected resistance of the foot skin, according to the literature [49]. Thus, it is reasonable to assume that the inter-electrode distance is possibly one of the most significant factors for signal saturation. However, in this work, this distance was kept at 0.5 mm to ensure the electrodes would not wrongly form a short-circuit, especially when integrating them within the socks’ fabric (i.e., fixing them to the sock/sewing over them).

From the subjects exhibiting at least partially measurable EDA signals, signal saturation does not seem to be related to anthropometric factors like age, gender, height, weight, and BMI, or body motion and foot temperature during the signal acquisition, although research suggests that tendency to sweat tends to decline with age [50].

Correlations obtained for filtered EDA signals are considered good when compared to similar research testing the same body locations. As an example, Payne et al. [49] observed hand and foot signals with a mean Pearson correlation of 0.53 and a range of [−0.07,0.91] for SCL in healthy individuals and using pre-gelled electrodes. Moreover, Kappeler-Setz et al. [11] reported the Pearson correlation between hand and foot SCL levels to be within [0.1,1.0] in limb moving conditions and, in some subjects, negative correlations were obtained when watching an action movie. In this same paper, the overall consensus for peaks within 4 s windows around reference hand peaks was 63% for 8 subjects. In our experiments, the mean value of 69% was obtained, which is slightly higher.

As for the time lags between nearby hand and foot peaks, a lag of 0.41±0.70 s was obtained, which is within the reported mean lag between hand and foot EDA signals of 1.3 s found in one paper [51]. Overall, the results from the EDA measurements suggest that it is feasible to use conductive textiles as EDA electrodes. Still, it is important to take into account the lower conductance levels at the foot to design a more suitable EDA sensor.

As for the PPG signals measured at the ankle, we obtained a sensitivity of 88.53±14.15 and a positive predictive value of 90.23±12.8, which were lower than those obtained from the finger signals, both in our research and in [46]. Moreover, the root mean squared error of the HR for the ankle was 17.40±14.80 BPM, which was considerably higher than 9.1±3.1 BPM, obtained from ankle measurements reported in [10]. It is known that weight, height, BMI, age, and gender can influence morphological features of the PPG signal [52]. In our analysis, sensitivities tended to increase with age, weight, and BMI and were higher for the male sub-group. We cannot conclude if the factors weight and BMI were associated with higher sensitivities because of increased contact pressure between the skin and the PPG sensor. Although the performance metrics were generally high for the PPG acquisitions at the ankle, the overall estimated HR had a clinically significant deviation.

Overall, available evidence suggests it is feasible to develop a smart sock for EDA and PPG acquisitions that would work reliably upon minimal body movement. Although these testing conditions do not represent all possible use cases of health monitoring in patients with neuropsychiatric disorders, they are relevant for discriminating health events (e.g., medical signs and symptoms) that do not necessarily co-occur with or constitute body movement. Two examples of such events would be non-convulsive seizures accompanied by autonomic dysregulation in patients with epilepsy, and manic and depressive episodes in patients with bipolar disorder.

Conductive lycra was chosen as the material for the EDA electrodes in this proof-of-concept study; this widely available material allowed for successful integration into the fabric substrate of conventional socks. However, with the availability of industrially-feasible techniques for producing conductive and stretchable fibers, it is worth considering the use of conductive polymers and intrinsically conductive yarns in future work [53]. This would be beneficial in determining the viability of scaling up the sock form factor, particularly considering its EDA-monitoring capabilities.

## 7. Conclusions

To the best of our knowledge, this is the first paper comparing EDA signals acquired at a reference body location and at the medial arch using textile electrodes integrated into a sock, in a thoroughly characterized population. Additionally, a PPG sensor was used to validate HR monitoring from the ankle region in the same sock. After testing the resulting sock for multimodal acquisition in multiple subjects, the obtained measurements were compared to similar experiments reported in the literature using the same performance metrics. The results suggest the proposed smart sock is able to effectively monitor EDA signals at the foot in some subjects, although others presented saturated EDA signals. For the latter subjects, signal saturation was possibly related to lower conductance values at the feet (i.e., <1 μS), for which an EDA sensor with a wider measurement range would be needed. As for the PPG acquisitions, by using an off-the-shelf pulse sensor at resting conditions, it was possible to obtain signals from which the HR could be successfully extracted, although with a clinically relevant HR estimation error. Future work should focus on experimental studies exploring variables such as skin tone, the effect of motion in the collected data, age, and pathological conditions.

## Figures and Tables

**Figure 1 sensors-23-00620-f001:**
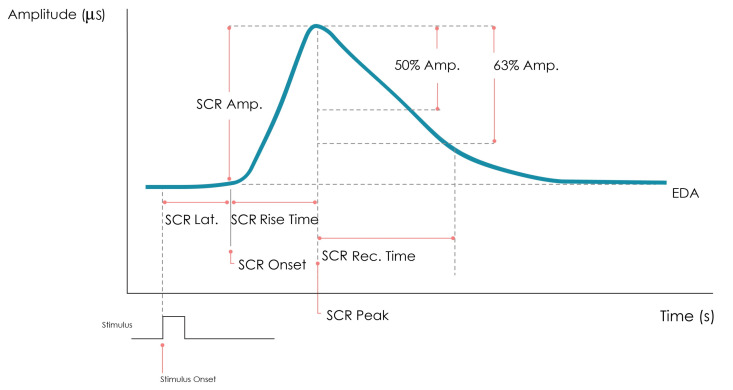
An illustrative example of a SCRs event.

**Figure 2 sensors-23-00620-f002:**
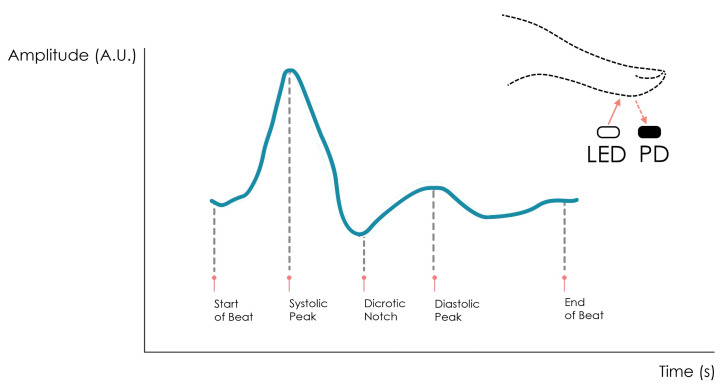
Illustrative example of a PPG signal obtained from a reflective-mode PPG sensor.

**Figure 3 sensors-23-00620-f003:**
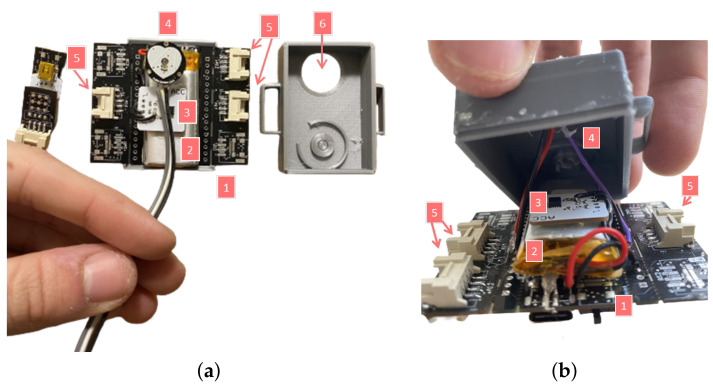
Inside view of the main unit, its main components, and prototyped enclosure. The various parts and components are labeled as follows: ScientISST CORE (1), small battery for power supply (2), triaxial accelerometer (3), PPG sensor (4), connectors where the remaining sensors are externally plugged (5), and the opening where the PPG sensor attaches internally within the enclosure (6). (**a**) Main unit open with sensors placed aside (top view). (**b**) Main unit with enclosure and wired sensors (front view).

**Figure 4 sensors-23-00620-f004:**
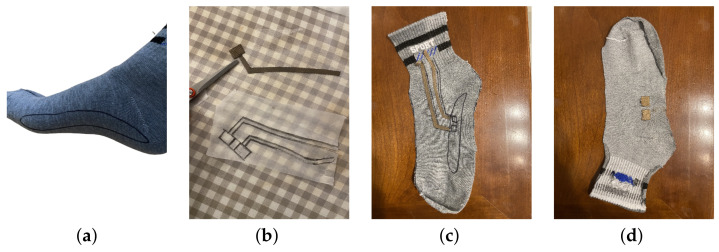
Main prototyping steps to obtain the sock with integrated textile EDA electrodes. (**a**) Outline of the medial arch on the sock. (**b**) Patch of conductive lycra cut from the drawn configuration of the two patches. (**c**) Outer side of the sock-half with the conductive segments. (**d**) Inner side of the sock-half with EDA electrodes.

**Figure 5 sensors-23-00620-f005:**
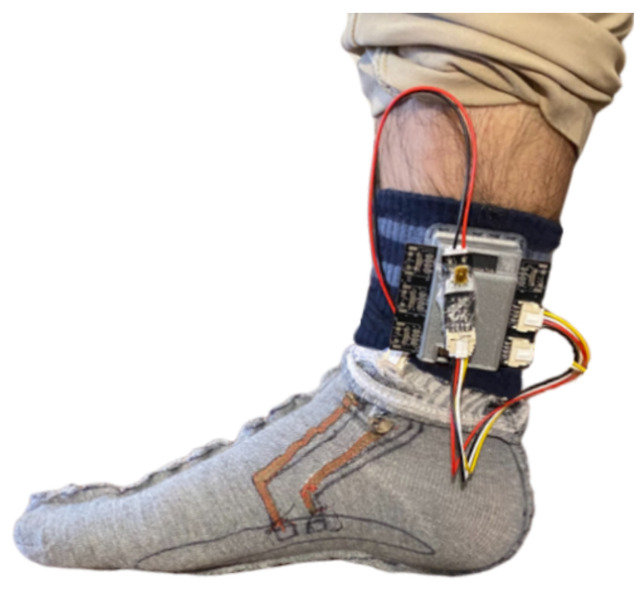
Prototyped proof-of-concept smart sock.

**Figure 6 sensors-23-00620-f006:**
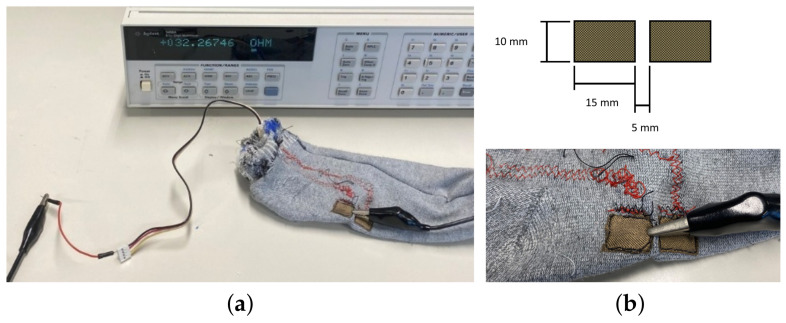
Measuring the electrical resistance across one of the conductive segments. Figure (**a**) shows the experimental setup, and Figure (**b**) is a zoomed version of the EDA electrodes, where one of the wires is connected, including a sketch of the electrodes’ dimensions (unstretched state).

**Figure 7 sensors-23-00620-f007:**
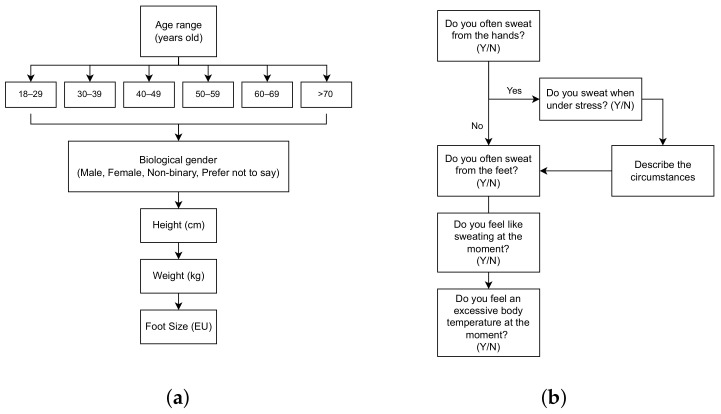
Survey for collecting subject data. (**a**) Questions for anthropometric data. (**b**) Questions for sweating tendency and sensorial perception.

**Figure 8 sensors-23-00620-f008:**
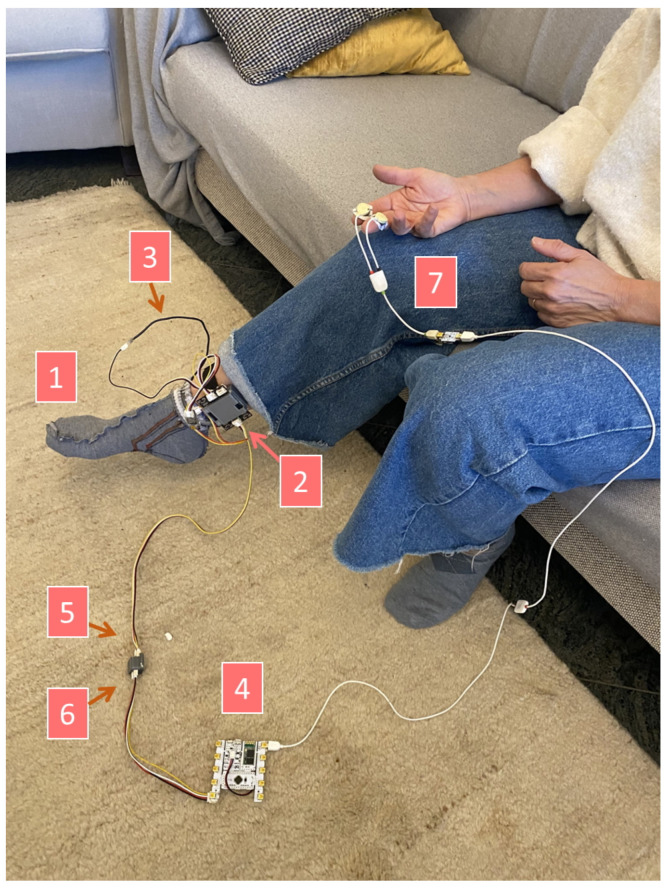
Experimental setup for evaluating the proposed system. The main parts are labeled with numbers: sensorized sock (1), main unit (2), temperature sensor (3), BITalino device (4), light sensor (5), LED (6), and EDA sensor and pre-gelled electrodes (7).

**Figure 9 sensors-23-00620-f009:**
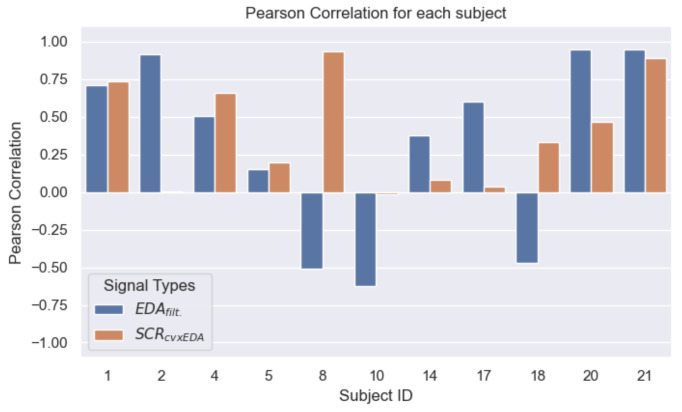
Distribution of the Pearson correlation values between measured hand and foot signals for each subject and type of pre-processed signals.

**Figure 10 sensors-23-00620-f010:**
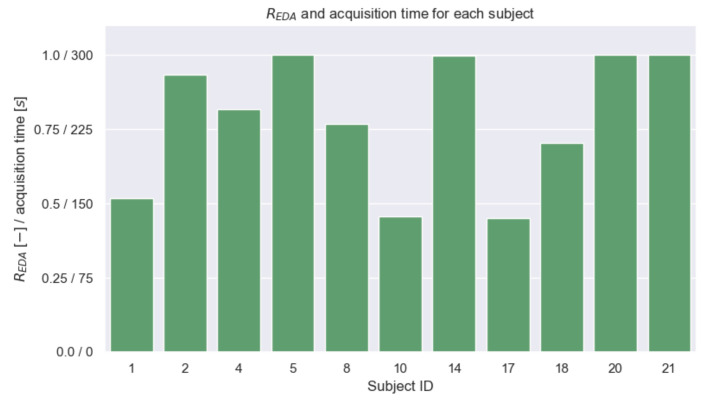
Distribution of the $R_{EDA}$ values for each subject, along with the resulting time lengths of the extracted segments of measurable EDA signals.

**Figure 11 sensors-23-00620-f011:**
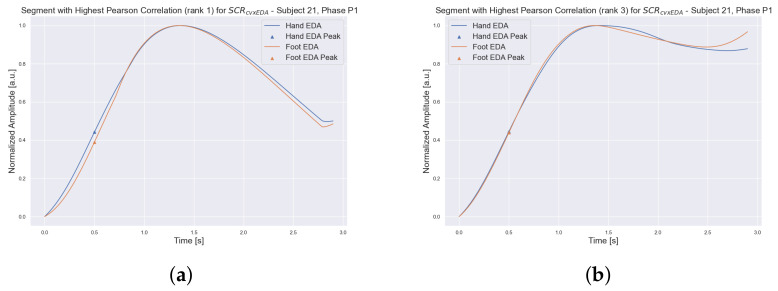
Hand and foot $SCR_{cvxEDA}$ segments corresponding to windows of co-occurring peaks. The displayed windows in Figure (**a**,**b**) have the highest and third highest Pearson correlation values for all segments $SCR_{cvxEDA}$ of co-occurring peaks, respectively.

**Figure 12 sensors-23-00620-f012:**
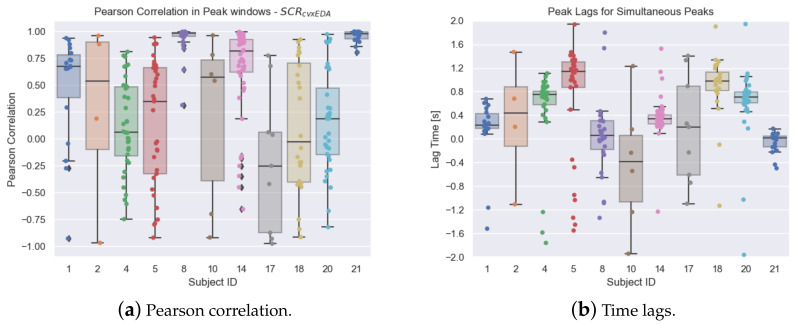
Analysis on the SCR component of the EDA signals measured at the hand and foot. (**a**) Distribution of the Pearson correlation values between extracted segments of hand and foot $SCR_{cvxEDA}$ within windows of co-occurring peaks. (**b**) Distribution of the time lags between nearby peaks. In both figures, the colorful dots represent the scattered data points of the boxplots.

**Table 1 sensors-23-00620-t001:** Summary of the resistance measurements: average, minimum, and maximum of all measurements for each conductive segment.

Conductive Segment/EDA Electrode	6 Measurements [Ω]	Min. [Ω]	Max. [Ω]	6 Measurements (Both Electrodes) [Ω]
Posterior	32.65±0.82	32.09	33.59	32.36±0.62
Anterior	32.06±0.18	31.92	32.26	

**Table 2 sensors-23-00620-t002:** Subject anthropometric parameters for the EDA acquisitions. The parameters are shown as median (min-max). For the age parameter, these values are expressed as the median age range (min–max). *p*-values were calculated using Mann–Whitney U test between the male and female groups.

Parameter	All Subjects, *n* = 19	Male Group, *n* = 11	Female Group, *n* = 8	*p*-Value
Age	[18–29] (18 to >70)	[30–39] (18 to 59)	[18–29] (18 to 69)	0.24
Height *, cm	168 (152–189)	174 (165–189)	163 (152–182)	0.02
Weight, kg	65 (48–100)	67.5 (59–100)	58 (48–88)	0.11
Body Mass Index (BMI), kg/m${}^{2}$	23 (16–38)	23 (21–28)	23 (16–38)	0.90
Foot Size *, EU	40 (35–45)	41 (40–45)	37 (35–43)	<0.01

* Parameters in which *p* values <0.05

**Table 3 sensors-23-00620-t003:** Subject anthropometric parameters for the PPG acquisitions. The parameters are shown as median (min–max). *p*-values were calculated using Mann–Whitney U test between the male and female groups.

Parameter	All Subjects, *n* = 15	Male Group, *n* = 5	Female Group, *n* = 10	*p*-Value
Age	[18–29] (18 to >70)	[18–29] (18 to >70)	[18–29] (18 to >70)	0.77
Height *, cm	164 (155–181)	173 (170–181)	163 (155–170)	<0.01
Weight *, kg	57 (50–95)	76 (74-95)	54.5 (50–84)	0.01
BMI, kg/m${}^{2}$	22 (19–32)	26 (23–31)	21 (19–32)	0.06
Foot Size *, EU	38 (35–43)	43 (42–43)	38 (35–40)	<0.01

* Parameters in which *p* values < 0.05

**Table 4 sensors-23-00620-t004:** Summary of the answers to the questions for assessing sweating tendency and sensorial perception for EDA acquisitions.

Question Nr.	Question	Answers	
		Yes	No
1	Do you often sweat from the hands?	7(37%)	12(63%)
2	Do you sweat when under stress?	8(42%)	11(58%)
3	Do you often sweat from the feet?	11(58%)	8(42%)
4	Do you feel like sweating at the moment?	4(21%)	15(79%)
5	Do you feel an excessive body temperature at the moment?	2(10%)	17(90%)

**Table 5 sensors-23-00620-t005:** PPG peak detection accuracy when using a band-pass filter with $f_{c_{L}}=0.97$ Hz and $f_{c_{H}}=11.12$ Hz in the filtering step. Each entry represents the metric value computed from all manually labeled (ground-truth) and automatically labeled peaks in all PPG signals (i.e., the total numbers of true positives, false positives, false negatives, and true negatives) and, in parenthesis, the mean and standard deviation of the values computed from individual segments (i.e., each segment from each subject).

Body Locations	Accuracy [%]	Positive Predictive Value [%]	Sensitivity [%]
Index Finger	99.46 (98.5±3.44)	99.64 (98.69±3.41 )	99.82 (99.81±0.33)
Ankle	87.85 (88.53±14.15)	90.69 (90.23±12.8)	96.56 (97.34±4.6)

**Table 6 sensors-23-00620-t006:** Candidate variables which resulted in statistically significant differences (*p*-value <0.05) in PPG peak detection sensitivity.

Candidate Variable (Stat. Test)	Statistic	*p*-Value
Age (Spearman Corr.)	0.56	0.02
Gender (Mann–Whitney U)	64.0	0.03
Weight (Pearson Corr.)	0.56	0.02
BMI (Pearson Corr.)	0.57	0.01

## Data Availability

Not applicable.

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
