# Peer review of "Feasibility of Electrodermal Activity and Photoplethysmography Data Acquisition at the Foot Using a Sock Form Factor"

_sensors, 2023, doi:10.3390/s23020620_

Round 1
Reviewer 1 Report
This paper presents a model for integrating sensors into a wearable structure for simultaneous monitoring of a person's EDA and pulse by a plethysmographic (PPG) method. The solution chosen by the authors is based on attaching the sensors to a special sock model, to which they add the electronic circuits for signal processing.
The paper has a well-chosen logical structure and presents in detail the stages of development of the subject addressed. The presentation of statistical data in figures and tables gives a more easily interpretable picture compared to the listings in the text (although the latter are also required).
However, it is necessary to revise the Conclusions chapter, because the authors' statements in lines 525-529 give the impression that the work is not actually finished, but that some detail issues, such as the use of an amplifier with adaptive gain control, still need to be resolved. Since the paper is not a multi-episode TV series, it should be obvious that the results associated with the current approach are definitive beyond any doubt. Also, the bibliographic reference [49] in the Conclusions must be removed.
Questions (related to the identification of possible anomalies in the electrode system implemented in the socks):
a) Is it possible to change the placement, shape and size of the electrodes to achieve a reliable electrode response in cases where the current design has failed?
b) Was more than one sock model used for the same person?
Other comments:
It is desirable to list the acronyms at the beginning of the paper to avoid multiple definitions. For example, Electrodermal Activity (EDA) is defined in both line 45 and line 64.
Row 164: replace “sensor with a bit resolution of 12 bit” with “sensor with 12 bit resolution”.
Rows 225-226: please check grammar.
What does n represent in equation (3)? Is this equation correct?
Rows 307-310: please rephrase the text to make it clearer.
Row 373: “the the following variables”.
Reviewer 2 Report
Comments for authors:
This research focused on the foot EDA signals using textile electrodes integrated in a sock, and a PPG sensor was employed to validate HR monitoring from the ankle region. The proposed method is able to effectively monitor EDA signals from the foot, and the measurement results were validated by recording the same signals from the hand. Although some interesting experimental results were obtained in this manuscript, it should be further improved. The below comments should be addressed.
1. The details of the data acquisition circuit should be schematically drawn in this paper (FIG 3).
2. The data of the electrical resistance across the conductive segments should be given in the manuscript.
3. The resistance depends on the dimension of the object. Thus, the effective sizes of the EDA electrode should be given in Fig 6.
4. How about the effect of body movement (such as walking or running) on the foot EDA signals?
5. Does the environmental temperature affect the signal acquisition?
6. Flexible sensors (such as fiber-like resistive sensors (ACS Applied Materials & Interfaces, 2018, 10(7): 6624-6635)) have been widely studied recently. Flexible sensors could be woven into textiles or clothes, which are the developing trend of wearable devices. Can this type of sensor be integrated into socks for EDA acquisition? The reviewers hope the authors to discuss this in the paper, along with refs.
Reviewer 3 Report
Major concerns
1. The main aspect of the study that requires improvement is the study design and how the controls were chosen. In addition to using the reference values/ground truth from measurements using wet electrodes and clinical-grade PPG sensors at the hand, reference values from the foot should also be included. For example, for EDA, since subjects were asked to be immobile, wet electrodes on the sole of the foot should have also been included. That way, the authors would be able to easily distinguish between physiologic variation at different anatomic sites and measurement error due to the textile electrodes. Similarly, PPG could have been measured from the toe using similar clinical-grade sensors so that anatomic variation could have been excluded.
2. Another major concern is that performance across different skin tones, especially for PPG, needs to be measured. PPG has been found in many studies to perform more poorly in those with darker skin. This should be tested in any new device for PPG.
3. Finally, the authors need to clarify what their intended use case is. The obvious utility for a sock-like form factor is for long-periods of at-home inconspicuous monitoring. However, what the authors tested was a very short time-frame with no movement and at the end of the discussion section, they state "would work reliable upon minimal body movement". How does this make sense for their stated use case? Without any sense of how this system holds up with normal movement such as walking, lying down, sitting, standing, how can we understand the significance of the results? Textile electrodes are notorious for needing consistent contact with the skin, which would definitely be disrupted with movement and daily activity. They also usually corrode with sweat exposure over time. They would also need to withstand some sort of washing or cleaning process. While it may not make sense to test all of this in one study, not including any movement at all makes it hard to interpret the significance of the results.
4. A large portion of the subjects had no measurable EDA signal using this system on the foot. This is a major problem that is casually mentioned at the beginning of the results section. A full EDA signal was only measurable in 3 out of 19 subjects, which is a major cause for concern and disagrees with the conclusion that this system can be used as a reliable measurement tool. Though I cannot say for sure, this is maybe due to poor contact of skin with the textile electrode and lack of hydration or moisture. Moisturizing the foot beforehand and using something like a compression sock might have yielded slightly better performance. Again, having an additional control at the same site with wet electrodes would allow for better assessment of why this might be.
5. Pearson correlation is a poor measure of agreement for a pulsatile signal (SCR) and unsurprisingly yields unimpressive values that get worse when discarding the flat non-pulsatile regions that factor in more. Something like a Jaccard index or just a simple confusion matrix would show the agreement better and not exclude false positives. Also the comparison would make more sense to be done with a control measured at the same site to exclude physiologic variation from anatomic site difference.
6. There is no discussion of the limitations of the study in the Discussion section.
Minor concerns
7. In the introduction, the justification for the sock-form factor and placement is that "more discreet" options are needed for patients compared to wristwatches. However, a wristwatch is sufficiently discreet for daily wear. A better justification would be something to do with data quality or optimal placement of electrodes on the sole of the foot etc.
8. At the end of the introduction, the authors state the ACC is a location-agnostic modality. They should clarify what they mean by that because technically, ACC taken from the extremities vs trunk can vary and are used for different purposes in specific cases.
9. At the end of Section 4.1, the authors should state what the functional battery life of their system is.
10. For Table 5, the fact that manually annotated peaks were used as the ground truth should be stated more explicitly
Reviewer 4 Report
In this article, a wearable smart sock was developed with an EDA (Electrodermal Activity) sensor with ergonomic textile electrodes and a PPG (Photoplethysmography) sensor in the form of an ankle bracelet.
The system developed by the authors uses a common sock with integrated EDA electrodes, which are connected to a data acquisition system located in an ankle bracelet (herein referred to as the main unit). This anklet contains the acquisition device, a PPG sensor and accelerometer. The main unit uses the ScientISST CORE signal acquisition board specially developed for biomedical applications, with built-in Bluetooth module for wireless transmission of the recorded signals. It has a controllable sampling rate, fs, of up to 16 kHz, although 1 kHz was used in the experiments in this work, and it has an Analog-to-Digital Converter (ADC) 12-bit resolution. This entire system is powered by a small battery. In addition, the main unit has a specially designed hole to fit the PPG sensor and EDA sensors and temperature sensors.
Conductive textiles used as EDA electrodes were anatomically mounted inside the foot for EDA acquisition. The socks were cut in the sagittal plane of the foot and assembled in the half that contains the region of the medial arch (two Lycra patches serving both for signal conduction and as electrodes, as needed). Table 1 shows the measured resistances with the values ​​of 32.65 ± 0.82 and 32.06 ± 0.18 Ω for the posterior and anterior electrodes, respectively. Furthermore, from all 6 measurements on each segment, the maximum difference in resistance observed was 1.67 Ω, which is approximately 5.16% of the average resistance of both electrodes, 32.36 Ω. Since the skin resistance accounts for a higher impedance to the electrode resistances, in addition to the contact resistance (ie, between the electrodes and the skin), the signals are within the measurable conductance range of the sensor. A total of 34 persons, 13 men and 17 women aged over 18 years were tested in the experiments. Among the persons, 19 were assigned to the EDA acquisition and 15 to the PPG acquisition. Anthropometric data are summarized in Tables 2 and 3 for EDA and PPG acquisitions, respectively. Anthropometric parameters such as height and foot size had significant differences between male and female subjects. Statistical data such as mean, median, variance, standard deviation, asymmetry, entropy, min, max, interquartile range and mean absolute deviation were calculated through Pearson's correlation between all these variables.
Most of the published papers on the use of PPG and EDA sensors have been carried out in the upper limbs, mainly hands and wrists. According to the authors this is first time that this kind of e-textile sensor is integrated in feet. In this sense, this paper proposes a new approach for PPG and EDA measurements using textile electrodes integrated into a sock, in a well-characterized population.
In summary it’s a well-written and organized paper that contains relevant information about the e-textile PPG and EDA sensors. Based on that, the manuscript has quality enough to appear in Sensors and should be accepted for publication in the present form.
Round 2
Reviewer 1 Report
Thank you for accepting the proposed changes!
Author Response
Thank you again for your previous thorough review. If no other change is required, I ask you to not answer this message, because the MDPI platform may consider the reply as a revision that still needs to be processed. Thank you!